# Polyimide-Coated Glass Microfiber as Polysulfide Perm-Selective Separator for High-Performance Lithium-Sulphur Batteries

**DOI:** 10.3390/nano9111612

**Published:** 2019-11-13

**Authors:** Mi-Jin Kim, Kwansoo Yang, Hui-Ju Kang, Hyun Jin Hwang, Jong Chan Won, Yun Ho Kim, Young-Si Jun

**Affiliations:** 1School of Chemical Engineering, Chonnam National University, 77 Yongbongro, Buk-gu, Gwangju 61186, Korea; mijinkim999@gmail.com (M.-J.K.); gmlwn120@gmail.com (H.-J.K.); wgguswls@gmail.com (H.J.H.); 2Advanced Materials Division, Korea Research Institute of Chemical Technology (KRICT), 141 Gajeongro, Yuseong-gu, Daejeon 34114, Korea; sky215@krict.re.kr (K.Y.); jcwon@krict.re.kr (J.C.W.); 3Advanced materials and chemical engineering, KRICT School, University of Science and Technology, Daejeon 34113, Korea

**Keywords:** lithium-sulphur battery, polysulfide shuttling, separator, glass fibers, polyimide

## Abstract

Although numerous research efforts have been made for the last two decades, the chronic problems of lithium-sulphur batteries (LSBs), i.e., polysulfide shuttling of active sulphur material and surface passivation of the lithium metal anode, still impede their practical application. In order to mitigate these issues, we utilized polyimide functionalized glass microfibers (PI-GF) as a functional separator. The water-soluble precursor enabled the formation of a homogenous thin coating on the surface of the glass microfiber (GF) membrane with the potential to scale and fine-tune: the PI-GF was prepared by simple dipping of commercial GF into an aqueous solution of poly(amic acid), (PAA), followed by thermal imidization. We found that a tiny amount of polyimide (PI) of 0.5 wt.% is more than enough to endow the GF separator with useful capabilities, both retarding polysulfide migration. Combined with a free-standing microporous carbon cloth-sulphur composite cathode, the PI-GF-based LSB cell exhibits a stable cycling over 120 cycles at a current density of 1 mA/cm^2^ and an areal sulphur loading of 2 mgS/cm^2^ with only a marginal capacity loss of 0.099%/cycle. This corresponds to an improvement in cycle stability by 200%, specific capacity by 16.4%, and capacity loss per cycle by 45% as compared to those of the cell without PI coating. Our study revealed that a simple but synergistic combination of porous carbon supporting material and functional separator enabled us to achieve high-performance LSBs, but could also pave the way for the development of practical LSBs using the commercially viable method without using complicated synthesis or harmful and expensive chemicals.

## 1. Introduction

Ever-growing demand on energy for electricity and transportation and abrupt climate change increases energy security and environmental concerns. Since energy accounts for about 60% of anthropogenic CO_2_ emission, urgent action to decarbonize the energy sector is needed [1]. In order to keep the average temperature rise well below 2 °C above the pre-industrial level, the net cumulative CO_2_ emission must be lowered from 1.5 to around 1 trillion tonnes [2]. This implies that energy dependence on fossil fuel combustion must immediately be lessened more than 80%. It is envisioned that renewable energies (REs) like solar and wind will achieve the target reduction of CO_2_ emissions. Indeed, a transition from fossil fuel to REs is currently well under way. For example, global energy investment on REs was increased by 17% to ~$270 billion in 2014. The capacity generation from REs was ~100,000 MW around the globe in 2014, corresponding to 9.1% of the world’s electricity [3].

The successful penetration of the REs into the energy sector requires their implementation with energy storage systems (ESSs) due to intermittency and location issues. Accordingly, we demand a battery technology that is safe, inexpensive, and long-lasting without performance loss as well as featuring high energy/power density [4]. Despite great success in portable electronics, penetration of the current state-of-the-art battery technology, i.e., lithium-ion batteries (LIBs), into this market is hampered by several limitations. They require further cost reduction (<$100/kWh) especially for cathode material based on Co, Ni, and Mn. In addition, they are not only reaching their energy density limit (~240 Wh/kg), but also have critical safety concerns of the thermal runaway [5,6]. Lithium–sulphur batteries (LSBs) with a theoretical energy density of 2600 Wh/kg and generating thermally-stable Li_2_S (m.p. > 900 °C) have been considered a promising alternative to LIBs for over two decades [7,8]. Sulphur has a theoretical specific capacity of 1672 mAh/g and is inexpensive ($0.02/g), earth-abundant, and environmentally friendly. What is more, a major source of its elemental form is oil refineries in which sulphur is a by-product, and refinery sulphur production is projected to increase with the US and Europe tightening environmental regulations and the crude oil from the underground becoming heavier and thus more sulphurous [9]. In general, LSBs consist of a metallic lithium anode and a sulphur cathode. The active material sulphur or related species undergoes multi-step reduction/oxidation during discharge/charge procedures based on the redox couple: 16Li + S_8_ <–> 8Li_2_S [7]. The reaction-based mechanism, generating two electrons per one lightweight sulphur atom, endows LSBs with a much higher specific capacity and energy density as compared to those of the intercalation-based LIBs, whose active material often provides a single electron generation and contains a high portion of dead-mass components [10].

Despite all of these advantages, the practical application of LSBs is yet impeded by several technical challenges, including: (1) inherently low electronic conductivity of charge/discharge products, i.e., sulphur and Li_2_S; (2) active material loss by dissolution of lithium polysulfide (Li_2_S_n_, *n* = 4–8) formed by the reaction between sulphur and lithium into electrolyte; and (3) the subsequent passivation of the lithium metal anode by the soluble polysulfides transferred from the cathode [11]. In order to mitigate these issues, much research effort has, so far, been given to the development of conductive additive or porous supporting material with the dimensions on a nanometer scale to improve the electronic/ionic conductivity of sulphur, organic/polymer/solid electrolytes with low solubility for polysulfides to minimize their dissolution from the cathode, additives therein to protect the surface of the lithium metal anode from Li_2_S passivation, anode active materials which hardly react with polysulfides, etc. [12,13,14,15,16,17,18,19,20]. The implication of these researches is that it is almost impossible for one strategy to completely prevent the dissolution and subsequent shuttling of the polysulfides without loss in either cycle stability or sulphur utilization efficiency. For example, the well-designed carbon cathode entails high polysulfide dissolution after repeated cycles as well as the high utilization efficiency (>90%) of sulphur close to the theoretical value. Also, the sulphur utilization in the presence of solid/polymer electrolyte or highly concentrated electrolyte gets lower than in the typical liquid electrolyte such as a mixture of 1,3-dioxolane and 1,2-dimethoxyethane (1:1, v/v), although these electrolytes retard the polysulfide dissolution to some extent. Polyolefin-based separators such as polypropylene (PP) and polyethylene (PE) are commonly used in LSBs and coated with various organic and inorganic materials, but they exhibit large thermal shrinkage at elevated temperatures and contain flammable components with the risk of ignition and flame in the case of overcharging or overheating [21,22]. Although a solid/polymer electrolyte which acts as a physical barrier for dissolved polysulfide have been widely investigated, degradation of electrochemical performance due to low ionic conductivity is inevitable in addition to the composite electrode suffering from high contact resistance at the electrode-electrolyte interface [23,24].

Almost all of the battery cell components must simultaneously be improved to cope with the challenges. Given that one strategy often counter-balances the advantage of the other, it is highly demanding to first investigate what might be a synergistic combination to achieve high-performance LSBs. To this end, we herein investigated the electrochemical behavior of porous carbon support toward sulphur in the presence of functional separator. It was our intention to create a synergy effect between the two different cell components in LSB cells where they were chosen to be either only electronically or ionically active such that they did not play overlapping roles. As such, the supporting material mainly improved the utilization of sulphur, while the separator retarded the polysulfide shuttling into the Li metal anode. Microporous carbon cloth (CC) was utilized as a sulphur supporting material without using additional conductive additives and polymer binders. The free-standing electrode infiltrated with polysulfides demonstrated a high performance of 1100 and 548 mAh/g corresponding to a sulphur utilization efficiency of 66 and 33% at an areal sulphur loading of 2 and 6 mg/cm^2^, respectively, and 0.3C [25]. The commercial glass microfiber (GF) separator with an open porous structure, which otherwise exhibits high electrolyte permeability and is prone to polysulfide shuttling of polysulfides, was rendered polysulfide-affinitive by introducing a homogeneous thin coating of polyimide (PI) in which the imide groups, two C=O groups linked to N, were well known to have high affinity to polysulfides [26]. The electrochemical behavior of CC was compared with or without PI coating on GF at an electrolyte-to-sulphur (E/S) ratio of >10 by using the galvanostatic cycling with potential limit (GCPL) technique and electrochemical impedance spectroscopy (EIS). The resulting CC and polyimide functionalized glass microfibers (PI-GF) before and after GCPL cycling were also analyzed by SEM, EDS, and TGA.

## 2. Materials and Methods

### 2.1. Materials

Pyromellitic dianhydride (PMDA) and m-tolidine (m-TB) were purchased from Sunlight Chem Inc. (Changzhou, China) and used after drying in a vacuum oven at 60 °C for 24 h. N-Methyl-2-pyrrolidone (NMP, 99.8%), and dimethylethanolamine (DMEA, 99.5%) were purchased from Sigma-Aldrich (Darmstadt, Germany) and used as received without further purification. The glass microfiber (GF/C grade) membrane (Whatman lnc., Buckingham, UK) pore size: 1.2 μm, thickness: 260 μm) was used as received.

### 2.2. Fabrication of PI Functionalized GF (PI-GF)

Polyamic acid (PAA), the precursor solution of polyimide, was synthesized by using PMDA and m-TB. First, m-TB (21.229 g, 0.1 mol) was dissolved in NMP (387.369 g) at 0 °C under a nitrogen atmosphere. After complete dissolution, PMDA (21.812 g, 0.1 mol) was gradually added. A viscous and yellow-colored PAA solution with a solid content of ~10 wt.% was acquired after stirring for 24 h. To obtain the PAA powder, the PAA solution was poured into deionized water, pulverized using a blender, filtered, and finally dried in a vacuum oven at 40 °C overnight. The aqueous PAA solution was then prepared by adding DMEA, the water dissolution agent, in water with a double amount of repeating units [27]. PI-GF was achieved by simply dipping the GF membrane into the as-prepared aqueous PAA solution. The immersed GF membrane was dried in a convection oven at 50 °C for 1 h and then heated in a vacuum oven at 350 °C for 2 h for thermal imidization.

### 2.3. Preparation of Electrolytes and Catholytes

The blank electrolyte is 1 M lithium bis(trifluoromethanesulfonyl) imide (LiTFSI, 99.95%, Sigma-Aldrich Inc., Darmstadt, Germany) dissolved in a 1:1 (v/v) mixture of 1,3-dioxolane (DOL, 99.8%, Sigma-Aldrich Inc., Darmstadt, Germany) and 1,2-dimethoxyethane (DME, 99.5%, Sigma-Aldrich Inc., Darmstadt, Germany) with 1 wt.% lithium nitrate (LiNO_3_, 99.99%, Sigma-Aldrich Inc., Darmstadt, Germany). To prepare the catholyte (1 M Li_2_S_6_), the sulphur source, elemental sulphur (Sigma-Aldrich, ≥99.5%) and Li_2_S (99.9%, Alfa-Aesar Inc., Seoul, Korea) were mixed in a blank electrolyte and heated at 50 °C overnight.

### 2.4. Cell Assembly

LSB cells were assembled using CR2032 coin cells in an Ar-filled glove box (below 0.1 ppm of O_2_ and H_2_O concentration). The catholyte was immersed into the commercial carbon cloth (CC, CH900-20, Kuractive, Kuraray Chemical Co., Ltd., Tokyo, Japan, 21 mg), to which the blank electrolyte (100 μL) was added to make the total amount of electrolyte 200 μL. Then, the PI-GF (or GF) separator with a diameter of 19 mm and the Li metal anode with a diameter of 10 mm and a thickness of 0.75 mm were sequentially placed on top of the CC cathode.

### 2.5. Characterization

Thermogravimetric analysis (TGA-50) was conducted under nitrogen in the temperature range of 25 and 900 °C at a heating rate of 10 °C/min (CDFC). Fourier transform infrared (FT-IR) spectra were recorded with a JACSO FT-IR 4100 spectrometer (JACSO Inc., Tokyo, Japan). Field emission scanning electron microscopy (FE-SEM) images were obtained using HITACHI SU-70 (HITACHI Inc., Fukuoka, Japan) in the Korea Basic Science Institute (KBSI). The cycle stability was evaluated in the potential range between 1.8 and 2.6 V vs. Li/Li^+^ with a WBCS 3000 multi-channel cycler (WonATech, Seoul, Korea). Electrochemical impedance spectroscopy (EIS) was measured from 250 kHz to 0.05 Hz by a VSP (Bio-Logic, Seyssinet-Pariset, France) potentiostat.

## 3. Results and Discussion

The GF membrane essentially consisted of borosilicate microfibers with a thickness in the range between 120 and 1900 nm (Figure 1a,b). Their random arrangement stacking with a thickness of 260 μm generated an open porous structure with a pore size of around 1.2 μm which was about 28 times bigger than that (0.043 μm) of the typical battery separator based on polypropylene (PP), i.e., Celgard 2400. In addition to the thermal stability as high as 550 °C (vs. ~90 °C for Celgard 2400), much higher than the thermal imidization temperature of PI, GF featured much higher electrolyte permeability than the Celgard membrane, which made it an ideal model separator (or substrate) to investigate the effect of the surface functional group, i.e., PI, on polysulfide shuttling without disturbance of physical trapping in the internal pore structure [26,28].

PI was in marked contrast to PP in that it had high thermal stability (>500 °C) and wettability to electrolyte solvent. PI was introduced into GF by simple dipping into aqueous PAA solution and thermal curing as shown in Figure 1c. The PAA loaded on GF was fully converted into PI by heating at 350 °C for 2 h, after which the resulting PI-GF with a pale yellow color indicative of PI formation could be obtained (Figure 1g inset). The amount and depth of color of PI on GF could gradually be modified by dipping GF into aqueous PAA solution of different concentrations (0.3, 0.5, and 1.0 wt.%). According to the concentration, these separators were referred to as PI-GF-*x* (*x* = 0.3, 0.5, and 1.0).

To enable the formation of the homogenous PI thin film on the hydrophilic surface of GF, we used water as a solvent for PAA instead of organic solvent to improve wettability and coating properties via hydrogen bonding [29,30]. In general, PAA has only limited solubility in water, while it dissolves well in polar aprotic organic solvents such as NMP, DMF, and DMAc [31]. Dissolution in water can be facilitated by reacting with the so-called water dissolution agents which are tertiary amines or alcohol amines such as dimethylethanolamine (DMEA) and trimethylamine. These amine compounds yield a salt complex with PAA via acid-amine reaction, substantially increasing solubility in water [27]. We found that the surface of GF indeed got wet faster with aqueous PAA solution than with organic PAA solution in NMP. As such, the PI coating resulting from aqueous solution was homogeneous and smooth, while that from the NMP solution was quite self-aggregated as shown in SEM images (Figure 1f). The surface morphology of GF remained almost intact before and after PI formation (PI-GF-0.5), meaning that the ultra-thin PI coating layer was homogeneously formed on the surface of GF thanks to the strong surface affinity of PAA to GF without affecting the chemical structure of silica. This holds true for PAA solution of ~3 wt.% (Figure 1d,e).

FT-IR confirms the formation of PI on GF: the weak absorption bands of the imide ring, consisting of carbonyl carbons bound to nitrogen, newly appeared at 725 (deformation), 1363 (C–N), and 1722 (C=O, sym.) cm^−1^ in the FT-IR spectrum of PI-GF-0.5 [32,33] (Figure 1g). The conversion of PAA into PI was successful as was revealed by the disappearance of the characteristic broad bands in the range from 3000 to 4000 cm^−1^ assigned to the N–H and O–H bonds of PAA. The other strong bands at 450, 787, and 1013 cm^−1^ were attributed to the bending vibration, symmetric stretching vibration, and asymmetric stretching vibration of Si–O–Si, respectively, of silica retained in borosilicate microfibers [34,35]. The content of PI in PI-GF-*x* was evaluated by heating PI-GF-*x* samples in a muffle furnace at 550 °C for 4 h under air. As expected, mass loss increased with increasing *x* from 0.3 to 1.0; PI-GF-0.3–1.0 samples showed mass losses of 2.0, 4.6, and 6.7 wt.%, respectively, which corresponded to the PI content in PI-GF-*x* (Figure 1h inset). This supported our assumption that the PI loading on GF could simply be fine-tuned with the initial concentration of aqueous PAA solution in the range between 0.3 and 1 wt.%. Note that the actual content of PI was probably lower by around 1 wt.% than the values obtained after the thermal decomposition in a muffle furnace; the borosilicate fibers also contributed to a mass loss of ~1 wt.% as a result of the removal of surface hydroxy groups (Figure 1h). Also, it was found that PI-GF guaranteed thermal stability up to 300 °C with a mass loss below 1 wt.%, above which PI-GF-0.5 showed an abrupt mass loss around 400 °C and continuous mass loss up to 700 °C, partly as a result of thermal decomposition of PI (Figure 1h).

Before the electrochemical characterization of PI-GF-*x* samples in LSB cells, we first assessed the adsorption capability of polysulfides by PI powders to see whether PI-GF-*x* provided sufficient heterogeneous adsorption sites (or PI content) for polysulfides. PI powders (50, 100, and 150 mg) prepared under the same conditions except using GF were mixed with 25 mM Li_2_S_6_ solution (1.6 mL) for 12 h under vigorous stirring in an Ar-filled glovebox until it reached equilibrium. Photo-images of the resulting supernatant showed that the catholyte solution of dark brown gradually become colorless with increasing the amount of PI powder and finally turned into a clear solution at 150 mg. Similarly, the absorption at both 350 and 450 nm assigned to S_6_^2-^ dissolved in DOL:DME (1:1, v/v) was substantially decreased with increasing PI powder in UV-Vis spectra of the supernatant [36,37,38]. With these results, it was roughly estimated that the adsorption capacity of PI powder for Li_2_S_6_ was around 0.4 μmol Li_2_S_6_/mg PI. Assuming a sulphur utilization efficiency of 70% and complete dissolution of the remaining 30% into the electrolyte at an areal sulphur loading of 2 mgS/cm^2^, we needed at least 12 mg PI, which was an order of magnitude more than the PI content in PI-GF-1. Although this meant that all PI-GF-*x* samples were obviously short of the surface adsorbents, it was very likely for the thin PI coating on GF to expose more carbonyl groups featuring high affinity to polysulfides as heterogeneous adsorption sites due to the nanostructure [39]. Indeed, it could be seen that the self-discharge of LSB cells at open-circuit voltage (OCV) was much reduced with PI-GF-*x* samples as compared to that with the Celgard 2400 (Figure 2d,e). We noted from the stable OCV values with only GF that the surface silanol groups of borosilicate microfibers were also valid for polysulfides adsorption and thereby self-discharge protection, which was in good agreement with previous reports proving the efficient trapping of polysulfides in porous silica [40,41]. However, it was exceptional that the PI-GF-1.0 sample showed rather poor self-discharge protection ability than other samples, including the bare GF. As will be discussed later, the PI content in PI-GF of more than 1 mg precluded ionic transfer especially in the presence of polysulfides. We thus limit our discussion to PI-GF-*x* samples at *x* ≤ 1.

Recently, we demonstrated that microporous carbon cloth (CC) with a small pore size of 1.2 nm, a high surface area of 1630 m^2^/g, and a high pore volume of 0.87 cm^3^/g was a decent sulphur-supporting material for high-performance, flexible LSBs [25]. Without using additional conductive additives, polymer binders, and porous supporting materials, the resulting LSB cells exhibited a specific capacity of 1100 and 548 mAh/g corresponding to a sulphur utilization efficiency of 66% and 33% at an areal sulphur loading of 2 and 6 mg/cm^2^, respectively, and 0.3C. In particular, a capacity loss of 0.137 and 0.129% per cycle was only marginal over 100 cycles due to the efficient physical trapping of polysulfides in micropores of ~1.2 nm. In the permanent presence of polysulfides of ~34% in electrolyte, this was, however, enabled partly by sacrificial LiNO_3_ additive capable of passivating the surface of the Li metal anode. The problem here was that such a protective SEI layer based on LiSO_x_ was temporary and would continuously degrade over cycles, eventually leading to cell failure [20,42]. It should be noted that most of the recent researches still rely on LiNO_3_ additives irrespective of their strategies. Additional trapping options for polysulfides are indispensable to avoid the same fate of LSB cells where it is important to minimize the detrimental effect of polysulfides shuttling on the Li metal anode if there is.

In order to investigate the effect of PI-GF-*x* on the electrochemical performance of CC, galvanostatic cycling with potential limitation (GCPL) and electrochemical impedance spectroscopy (EIS) measurements were conducted at an areal sulphur loading of 2 mgS/cm^2^ and an areal current density of 1.0 mA/cm^2^, corresponding to 0.3C, in the potential range between 1.8 and 2.6 V vs. Li/Li^+^. The E/S ratio was set to be ~60 by using 1 M Li_2_S_6_ catholyte solution as a sulphur source, under which condition the electrolyte was considered a dilute solution. The low concentration gradient made it difficult for PI-GF-*x* to drive surface adsorption of polysulfides toward equilibrium. We also used the tough condition to exaggerate the polysulfide shuttling in LSB cells as the dilute electrolyte drove more dissolution of polysulfides from the cathode. These cells were rested at OCV for 12 h before the GCPL test to drive self-discharge reaction if there was any, and then first charged to the upper potential limit in order to electrochemically infiltrate sulphur into CC. The first charge curve was removed from the GCPL profiles in order to avoid confusion unless otherwise noted.

Figure 3a shows a capacity vs. cycle number graph of PI-GF-*x* samples. It is clear that PI-GF-0.5 outperformed other samples including GF in terms of capacity generation and cycle stability; PI-GF-0.5 generated a maximum capacity of 1328 mAh/g corresponding to a sulphur utilization efficiency of 79.5% with a high Coulombic efficiency of 99.12% and a marginal capacity loss of 0.099%/cycle over 120 cycles, while other samples showed a lower capacity generation by 10% and suffered from a sudden cell failure in 60–70 cycles. The CC cathode combined with PI-GF-0.5, as expected, still performed in the same manner as with the Celgard 2400 or GF (Figure 3b). The GCPL profiles of PI-GF-0.5 showed two typical plateaus at 2.3 and 2.0 V vs. Li/Li^+^ during discharge corresponding to reduction reactions of sulphur to high (Li_2_S*_x_*, 4 ≤ *x* ≤ 8) and low (Li_2_S*_x_*, 1 ≤ *x* < 4) polysulfides/lithium sulfide, respectively, followed by a gradually sloping charge profile toward the upper potential limit attributed to the subsequent oxidation reactions of lithium sulfide, and polysulfides to elemental sulphur [25]. For PI-GF-0.5, the capacity ratio of low (700–800 mAh/g) to high (550 mAh/g) polysulfides remained at ~1.45, which was much lower than the theoretical value (3). We attributed this to the microporous CC which is efficient at reactions of soluble high polysulfides responsible for shuttling, but poor at those of insoluble low polysulfides causing pore-blockage [25,43]. The PI-GF-0.5 contributed mainly to the improvement in the former by retarding polysulfides shuttling and thereby keeping the high polysulfides in close proximity to the reaction sites on CC, which was probably why PI-GF-0.5 showed no sign of shuttling during charge, i.e., charge profile drooping or fluctuation without reaching the upper potential limit. We do not exclude the fact that PI is an electrochemically active material for lithium ions in the potential range between 1.9 and 2.3 V vs. Li/Li^+^ well-enveloped in the lower and upper potential limit of GCPL measurement either [39]. It is, however, not only almost impossible to precisely determine the mass of PI taking part in the electrochemical reaction, but also highly unlikely that such a reaction consistently happens at a remote mode without establishing intimate contact with the current collector.

The role of PI-GF-0.5 in improving the electrochemical performance of LSB cells was investigated in more detail. It is well-known that polar polymers containing oxygen or nitrogen atoms in polymeric separators such as polyacrylonitrile, poly(methyl methacrylate) facilitate the lithium-ion transfer and distribution and thereby improves the performance of lithium metal anode in carbonate-based electrolyte due to their high affinity [44]. We assumed that the same is true for PI in ether-based electrolytes. Pyridinic nitrogen atoms in PI, which act as lithiophilic sites, has been reported to regulate the cation pathway and their nucleation [45]. Also, it is expected to reduce the overpotential required for lithium plating/stripping. Superior cycle stability and rate capability of PI-GF-0.5 to GF in lithium plating/stripping reactions using 1M LiTFSI solution dissolved in DOL:DME (1:1, v/v) were confirmed. During the cycle stability test of 1700 h (500 cycles) at a constant current density of 0.35 mA/cm^2^, the overpotential remained as low as 5 mV except for the initial fluctuation in PI-GF-0.5 during conditioning (Figure 4a). In the following rate capability test, the overpotential only slightly increased by 0.3 mV with increasing current density from 0.35 to 2.8 mA/cm^2^ and again remained stable even at 2.8 mA/cm^2^ (Figure 4b).

Despite the superior performance of the lithium metal anode, PI-GF-0.5-based LSB cells showed a slightly lower rate capability than those of GF-based ones. This means that the polysulfides adsorption affected the lithium-ion transfer through PI-GF-0.5 [46] (Figure 4c,d). In order to clarify the effect of polysulfide adsorption on the ionic conductivity of PI-GF-0.5, EIS measurement was conducted using Al/separator-electrolyte/Al symmetric cells. The ionic conductivity (σ) was calculated as follows:(1)$$\sigma =\frac{d}{R_{b}A},$$
where *d* is the thickness of the separator, *R_b_* is the intersection of the Nyquist plot with the Z-axis at the highest frequency, indicating the bulk electrolyte resistance, and *A* is the contact area between electrode and separator [47,48]. In the absence of the polysulfides, the ionic conductivity of PI-GF-*x* samples gradually increased with increasing the PI content, even 1.36 times higher than that of GF, matching well with the results of lithium plating/stripping reactions (Figure 5a,b). On the contrary, the ionic conductivity decreased with the PI content in the presence of polysulfides. The effect was especially pronounced at *x* = 1, where the high content of polysulfide-philic PI allowed adsorption of polysulfides to a large extent, substantially reducing the ionic conductivity and thus leading to sudden LSB cell failure of PI-GF-1. The same interpretation applies to the initial bulk cell resistance of PI-GF-*x*-based LSBs at an areal sulphur loading of 2 mgS/cm^2^ (Figure 5c). For example, the bulk resistance and charge transfer resistance noticeably increased for PI-GF-1 due to the high loading of polysulfides on its surface.

Photo and SEM images of the cycled electrodes and separators help us roughly qualify/quantify and visualize the mode of surface adsorption of polysulfides. The LSB cells using GF, PI-GF-0.5, and PI-GF-1 were disassembled after charge/discharge cycles at an areal sulphur loading of 2 mgS/cm^2^ and their changes in surface morphology, texture, and color were observed. Judging from the color change of the cycled separator samples, the polysulfide shuttling clearly happened in LSBs cells even at such low areal sulphur loading (Figure 6a–c). We assume from the utilization efficiency in GCPL experiments that around 20–30% of total sulphur added was dissolved into the electrolyte solution. GF and PI-GF-*x* turned yellow and green with polysulfides, respectively, where the latter agreed well with what has been seen in previous reports [49]. It was found in photo images of PI-GF-*x* that the higher the PI content, the darker the green. Similarly, we found from SEM images that the formation of particles and thin coating layers in PI-GF-*x* got more significant with increasing PI content. The particles filled the voids between the fibers of GF. In addition, the thin coating layers of polysulfides were created at the corner and completely covered the space by cohesive forces [50]. This is probably why the ionic conductivity of PI-GF-1-based LSB cells significantly reduced.

TGA results of the cycled PI-GF-0.5 show that the weight loss of about 7% and 12% was centered at around 100 °C and 450 °C, respectively. The latter is mainly attributed to the adsorbed polysulfide in PI-GF-0.5. Given that sublimation of elemental sulphur occurs mainly at < 300 °C, high evaporation temperature suggests that polysulfides bind strongly to PI-GF-0.5 [51]. EDS mapping and quantification results reveal that sulphur species were indeed well-distributed in the separator samples after DME washing. The PI-GF-0.5 contained about 10% less sulphur content compared to GF, probably indicating that the thin polysulfide layers created between fibers prevented the dissolved polysulfide from penetrating deep into the PI-GF-0.5.

## 4. Conclusions

To conclude, we demonstrated a synergistic combination of porous carbon cloth (CC) with a polyimide coated glass microfiber (PI-GF) for high-performance LSBs. CC and PI-GF function as a sulphur-supporting material and a polysulfide perm-selective separator, respectively, in which PI-GF helps polysulfides kept in close proximity to the CC by forming polysulfide films supported by microfibers via cohesive force. The degree of film formation is determined by the content of PI around 0.5 wt%, which requires delicate control. This was enabled by the aqueous PAA-solution-based synthesis of homogeneous thin PI coating on GF with the potential to scale and fine-tune. As a result, CC combined with PI-GF exhibited a sulphur utilization efficiency of 79.5% with a marginal capacity loss of 0.099 %/cycle over 120 cycles.

## Figures and Tables

**Figure 1 nanomaterials-09-01612-f001:**
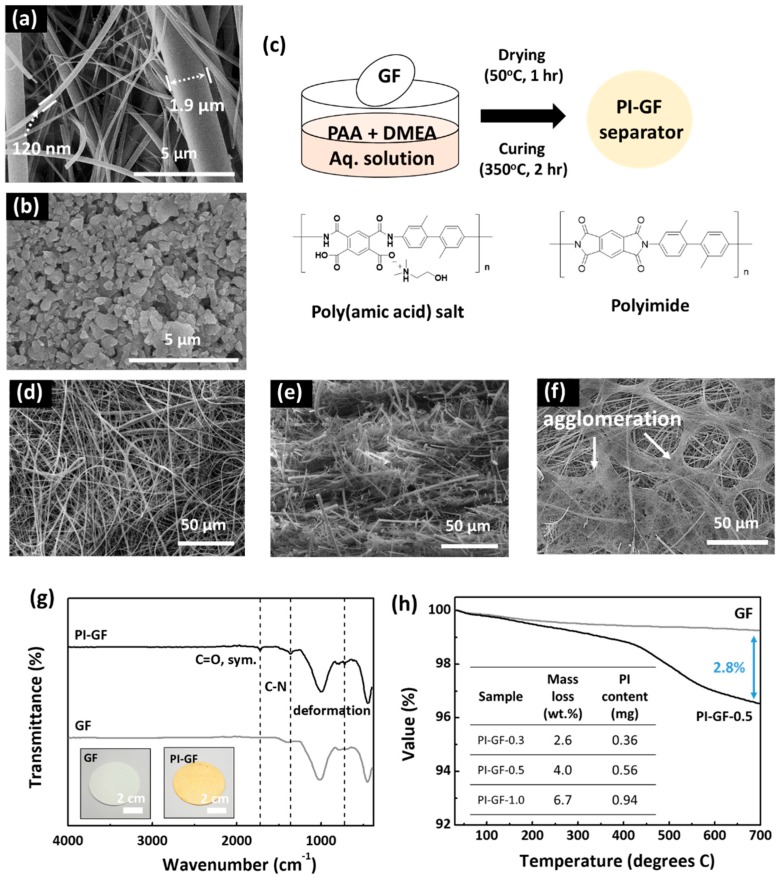
SEM images of (**a**) glass microfiber (GF); (**b**) polypropylene (PP); (**c**) the schematic diagram of the fabrication process for polyimide functionalized glass microfibers (PI-GF) separator and molecular structures of polyamic acid (PAA) salt and polyimide (PI); SEM images of (**d**,**e**) PI-GF-0.5 and (**f**) PI-GF-3 coated with organic (NMP) solutions; (**g**) Fourier transform infrared (FT-IR) spectra and (**h**) TGA curves of GF and PI-GF-0.5; (**g**, inset) photo images of GF and PI-GF; and (**h**, inset) mass loss and PI content in PI-GF-*x*.

**Figure 2 nanomaterials-09-01612-f002:**
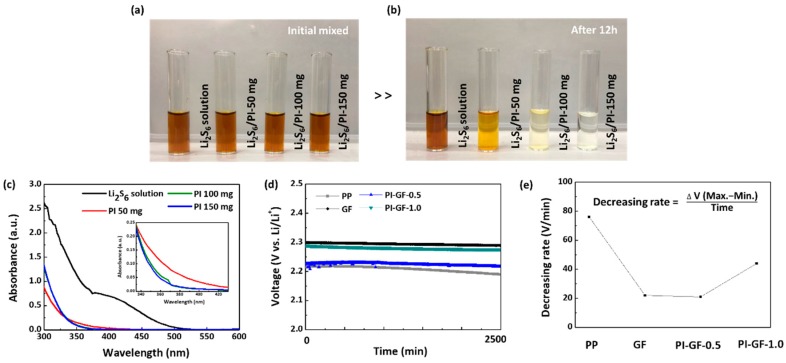
Adsorption test of polyimide powder in 25 mM Li_2_S_6_ solution: Photo-images of the supernatant of solutions after (**a**) initial mixed and (**b**) 12 h; (**c**) UV-vis spectra of the supernatant solutions; (**d**) self-discharge of lithium-sulphur batteries (LSBs) at open-circuit voltage (OCV); and (**e**) the corresponding decreasing rate.

**Figure 3 nanomaterials-09-01612-f003:**
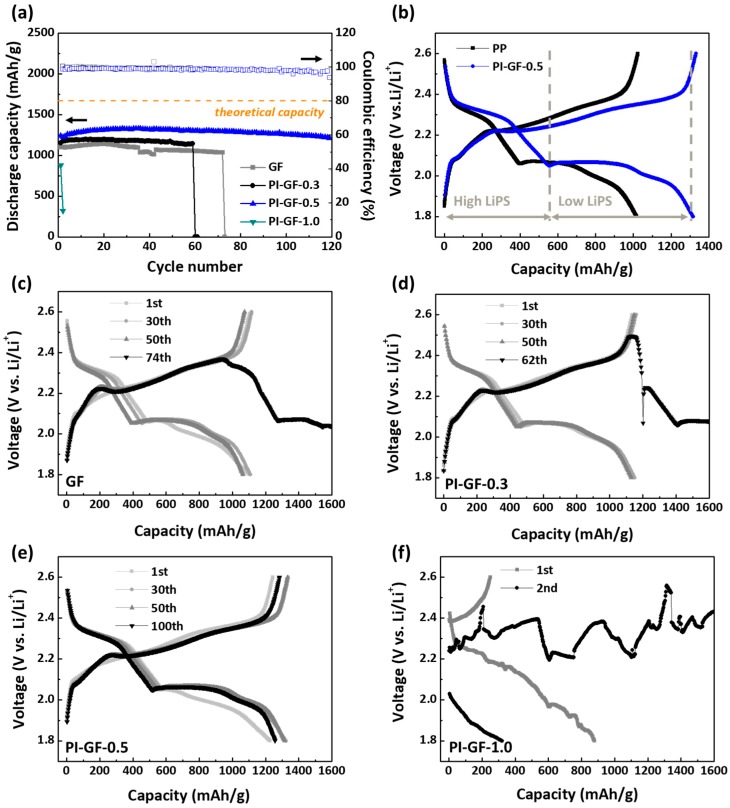
(**a**) Comparison of electrochemical performance depending on separators at 2 mgS/cm^2^ and 0.3C. (**b**–**f**) Charge/discharge profiles of different separators: (**b**) PP, (**c**) GF, (**d**) PI-GF-0.3, (**e**) PI-GF-0.5, and (**f**) PI-GF-1.0.

**Figure 4 nanomaterials-09-01612-f004:**
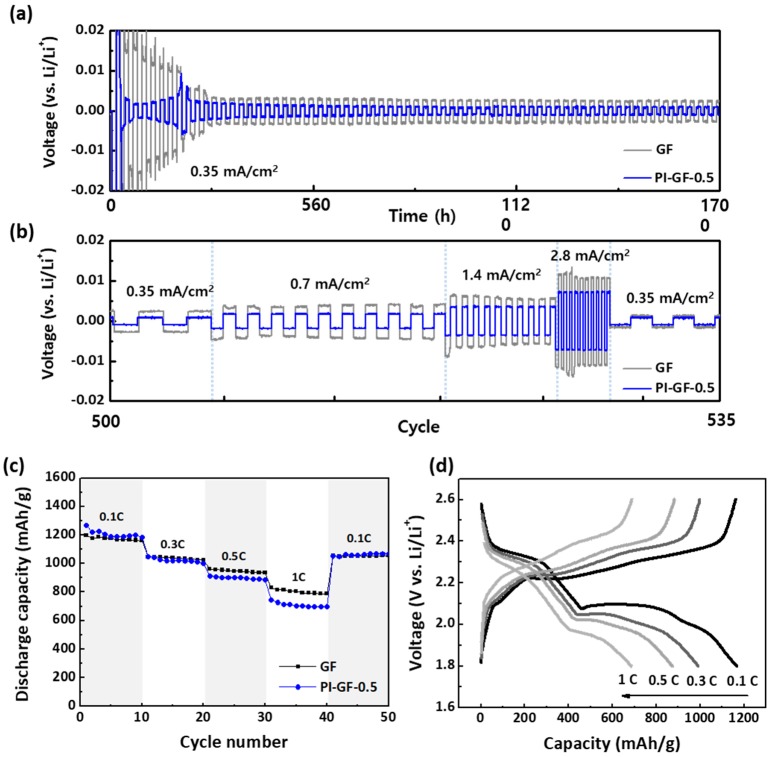
(**a**) Lithium stripping/plating performance in Li/Li symmetric cells with GF and PI-GF-0.5 separators: (**a**) cycle stability of 1700 h (500 cycles) at a constant current density of 0.35 mA/cm^2^ and (**b**) rate capability in the current density from 0.35 to 2.8 mA/cm^2^. (**c**) Rate capability performance of LSBs using the GF and PI-GF-0.5 separators at 2 mgS/cm^2^, (**d**) the corresponding charge/discharge profile of PI-GF-0.5.

**Figure 5 nanomaterials-09-01612-f005:**
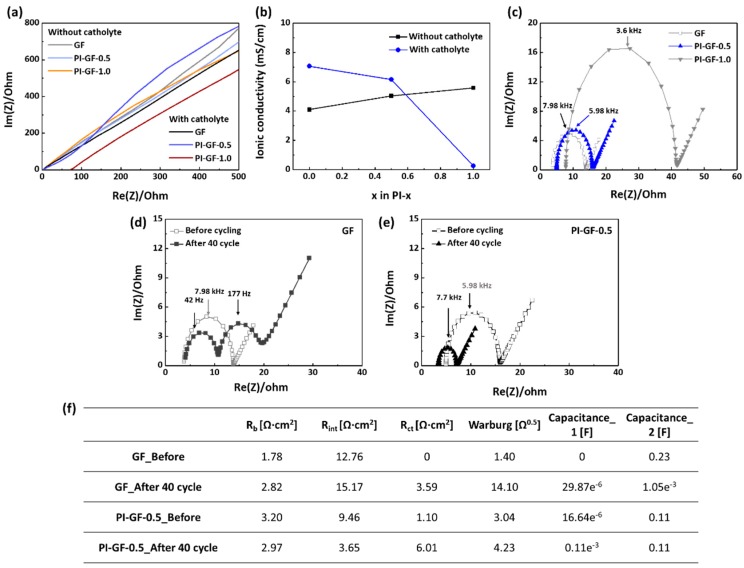
(**a**) Electrochemical impedance spectroscopy (EIS) Nyquist plot for Al/separator–electrolyte/Al symmetric cells and (**b**) corresponding ionic conductivity without and with 1 M Li_2_S_6_ solution, (**c**–**e**) EIS Nyquist plot for LSBs under the GF, PI-0.5, and PI-1 separators before and after cycling, and (**f**) corresponding fitted capacitance and resistance values.

**Figure 6 nanomaterials-09-01612-f006:**
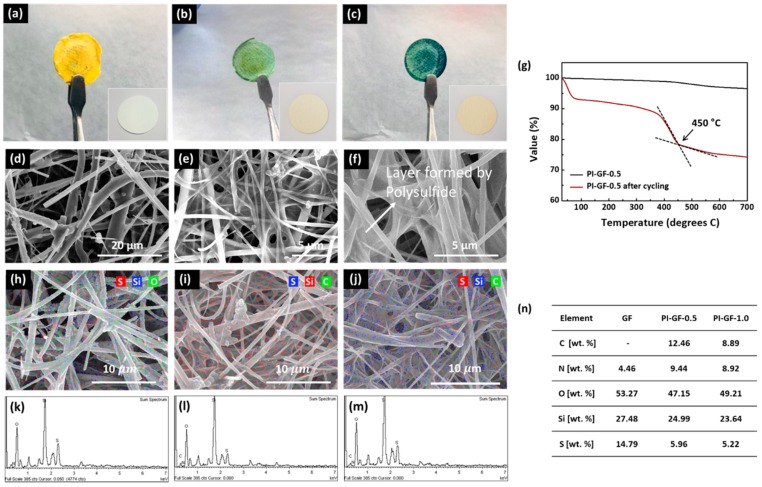
(**a**–**c**) Photo images and (**d**–**f**) SEM images: (**a**,**d**) GF, (**b**,**e**) PI-0.5, and (**c**,**f**) PI-1 separators after cycling, (**a**–**c**, inset) separators before cycling. (**g**) TGA curves of PI-GF-0.5 before and after cycling, (**h**–**n**) EDS mapping and analysis: (**h**,**k**) GF, (**i**,**l**) PI-GF-0.5, (**j**,**m**) PI-GF-1.0, and (**n**) EDS element analysis of separators.

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
