# Peer review of "Polyimide-Coated Glass Microfiber as Polysulfide Perm-Selective Separator for High-Performance Lithium-Sulphur Batteries"

_nanomaterials, 2019, doi:10.3390/nano9111612_

Round 1

Reviewer 1 Report

In this work, the authors improve substantially the lithium-sulfur batteries by preparing and using a functional separator, namely polyimide functionalized glass microfibers. The synthesis of the polyimide and the fabrication process is fairly simple and the results regarding the cycle stability and the specific capacity seem to be important and a high sulfur utilization efficiency is achieved.

Comments:

line 63: based on the redox couple: 16Li + S8 <-> 8Li2S  "   152 Figure 1 can be simplified. Photos of the PI-GF are not necessary, same for GF and PI-GF. SEM images of PP compared to GF have to be discussed. The other parts of this figure have to be diwscussed. the reaction of PAA with a tertiary amine should be included in Figure 1  as actually  it is the salt that will be coated, There are many sentences that are too long (for example line 26-31, 101-105 etc.) and have to be splitted for more clarity. I suggest the use of the more common spelling: sulphur, instead of the more American sulfur.   

I suggested to the authors to shorten their sentences for more clarity.

Author Response

Attached please find our responses to the Reviewer 1's comments.

Reviewer 2 Report

Authors carefully studied the ployimide coated glass microfibers as polysulfide seperator and they showed  improved electrochemical properties

Following below minor revisions are needed and clarified

Authors repeated cycling studies of Fig.3b , sometimes depending on cell fabrication we can expect this type cycling behavior, my suggestion under identical current density and mass , please repeat the galvanostic cycling studies and see the reproducibilty Fig.5C, please mention the frequency limits in the impedance  plots and also change y-axis to -ve  and also mention what what impedance studies are taken , it would have nice authors shown before and after cycling the impedance plots please included the fitted capacitance  and resisatnce values Introduction section to be improved by further by  batteries reviews ex : chemical reviews  113(2013)5364

Author Response

Attached please find our responses to the Reviewer 2's comments. 
